# Integrated Transcriptomics, Metabolomics, and Lipidomics Profiling in Rat Lung, Blood, and Serum for Assessment of Laser Printer-Emitted Nanoparticle Inhalation Exposure-Induced Disease Risks

**DOI:** 10.3390/ijms20246348

**Published:** 2019-12-16

**Authors:** Nancy Lan Guo, Tuang Yeow Poh, Sandra Pirela, Mariana T. Farcas, Sanjay H. Chotirmall, Wai Kin Tham, Sunil S. Adav, Qing Ye, Yongyue Wei, Sipeng Shen, David C. Christiani, Kee Woei Ng, Treye Thomas, Yong Qian, Philip Demokritou

**Affiliations:** 1West Virginia University Cancer Institute/School of Public Health, West Virginia University, Morgantown, WV 26506, USA; qiye@mix.wvu.edu; 2Lee Kong Chian School of Medicine, Nanyang Technological University, Singapore 308232, Singapore; poh_tuang_yeow@ntu.edu.sg (T.Y.P.); schotirmall@ntu.edu.sg (S.H.C.); sshen@hsph.harvard.edu (S.S.); dchris@hsph.harvard.edu (D.C.C.); 3Center for Nanotechnology and Nanotoxicology, Department of Environmental Health, T. H. Chan School of Public Health, Harvard University, Boston, MA 02115, USA; svp097@mail.harvard.edu (S.P.); kwng@ntu.edu.sg (K.W.N.); pdemokri@hsph.harvard.edu (P.D.); 4Pathology and Physiology Research Branch, Health Effects Laboratory Division, National Institute for Occupational Safety and Health, Morgantown, WV 26505, USA; woe7@cdc.gov (M.T.F.); yaq2@cdc.gov (Y.Q.); 5Singapore Phenome Centre, Lee Kong Chian School of Medicine, Nanyang Technological University, Singapore 636921, Singapore; tham_wk@ntu.edu.sg (W.K.T.); ssadav@ntu.edu.sg (S.S.A.); 6Key Lab for Modern Toxicology, Department of Epidemiology and Biostatistics and Ministry of Education (MOE), School of Public Health, Nanjing Medical University, Nanjing 210029, China; ywei@njmu.edu.cn; 7School of Materials Science and Engineering, Nanyang Technological University, Singapore 639798, Singapore; 8Environmental Chemistry and Materials Centre, Nanyang Environment & Water Research Institute, Singapore 637141, Singapore; 9Office of Hazard Identification and Reduction, U.S. Consumer Product Safety Commission, Rockville, MD 20814, USA; tthomas@cpsc.gov

**Keywords:** printer emitted nanoparticles, inhalation, transcriptomics, metabolomics, lipidomics, biomarkers, nanotoxicity

## Abstract

Laser printer-emitted nanoparticles (PEPs) generated from toners during printing represent one of the most common types of life cycle released particulate matter from nano-enabled products. Toxicological assessment of PEPs is therefore important for occupational and consumer health protection. Our group recently reported exposure to PEPs induces adverse cardiovascular responses including hypertension and arrythmia via monitoring left ventricular pressure and electrocardiogram in rats. This study employed genome-wide mRNA and miRNA profiling in rat lung and blood integrated with metabolomics and lipidomics profiling in rat serum to identify biomarkers for assessing PEPs-induced disease risks. Whole-body inhalation of PEPs perturbed transcriptional activities associated with cardiovascular dysfunction, metabolic syndrome, and neural disorders at every observed time point in both rat lung and blood during the 21 days of exposure. Furthermore, the systematic analysis revealed PEPs-induced transcriptomic changes linking to other disease risks in rats, including diabetes, congenital defects, auto-recessive disorders, physical deformation, and carcinogenesis. The results were also confirmed with global metabolomics profiling in rat serum. Among the validated metabolites and lipids, linoleic acid, arachidonic acid, docosahexanoic acid, and histidine showed significant variation in PEPs-exposed rat serum. Overall, the identified PEPs-induced dysregulated genes, molecular pathways and functions, and miRNA-mediated transcriptional activities provide important insights into the disease mechanisms. The discovered important mRNAs, miRNAs, lipids and metabolites may serve as candidate biomarkers for future occupational and medical surveillance studies. To the best of our knowledge, this is the first study systematically integrating in vivo, transcriptomics, metabolomics, and lipidomics to assess PEPs inhalation exposure-induced disease risks using a rat model.

## 1. Introduction

The unique physicochemical properties of engineered nanomaterials (ENMs) are being exploited for use in a growing variety of commercial nano-enabled products (NEPs), including electronics, cosmetics, agriculture and food, structural materials, as well as a wide variety of products for medical applications [1,2,3,4]. Numerous in vitro and in vivo studies have investigated the possible adverse effects of inhalation exposure to pristine ENMs during synthesis and handling [5,6,7,8]. However, human exposure is not limited to pristine ENMs, but to a wide variety of particles released from NEPs across their life cycle, including consumer use and disposal [9,10,11,12]. Indeed, the potential for exposure from such life cycle particulate matter (LCPM) may exceed that of pristine engineered nanoparticles (ENPs) [13]. All pristine materials are ENPs. LCPM is the term we introduced to explain that pristine ENPs across their cycle and value chain can be released in the air but their properties differ due to their transformations. For instance, an ENP which will be used in a nano-enabled paint will be released during sanding and drilling, but the released LCPM will have different physico-chemical properties compared to pristine ENP used in the paint. The same applies to the pristine metallic nanoparticles used to formulate the toner of laser printers. When those ENPs are released in the air they will have completely different physico-chemical properties and will include in their surface organic compounds that are not there for the pristine ENPs. Moreover, the toxicological profiles of LCPM may differ greatly from those of pristine ENPs [14,15]. Despite the potential for LCPM exposure, most nanotoxicology studies have focused on pristine ENPs and thus, toxicological evaluation of LCPM has been limited. The National Institute for Occupational Safety and Health (NIOSH) and the Environmental Protection Agency (EPA) have, therefore, recommended life cycle analyses in their nanotechnology research programs [16,17]. Most importantly, we need to develop integrated methodologies that can link real world LCPM exposures to toxicology and diseases.

The rate at which production of new ENPs and incorporation into NEPs occurred has far exceeded our ability to test all ENPs and LCPM released over the NEP life cycle using in vivo animal studies. This study will focus on a specific real-world LCPM: laser printer-emitted particles (PEPs) generated from nano-enabled toners which has been studied extensively by the authors and others over the last decade [18,19]. In US alone, 23 million printers are produced annually, and over 160,000 workers are employed in copier centers [20,21,22,23,24,25,26,27,28]. The new generation of toners contain ENPs, such as metals and metal oxides, and that PEPs with mode sizes of 49–208 nm are released during printing at concentrations of up to 1.3 million particles/cm^3^ [15]. There is growing evidence from both in vitro cellular and in vivo animal studies of significant bio-activity of PEPs, which points to potential adverse health effects from inhalation exposure [15,28,29,30,31,32,33,34,35,36]. Our group recently suggested exposure to PEPs increases cardiovascular risk by impairing ventricular performance and inducing hypertension and arrhythmia [37]. Currently, there are no available biomarkers for assessing adverse cardiovascular responses induced by PEPs exposure in susceptible individuals with long-term exposure.

Genomic/transcriptomic analysis has become a routine approach in biomedical research and has increasing applications in toxicology [38,39]. However, this is not the case in nanotoxicology research. Gene expression analysis could aid mechanistic studies by evaluating the relevance of signaling pathways representing significantly perturbed genes for treatment interventions [40,41,42]. Since miRNAs are more stable than mRNA, and can be obtained from minimally-invasive biological samples, such as blood, they can provide more insight into identifying valuable diagnostic and predictive biomarkers [43,44]. Integrated analysis of miRNA and mRNA could reveal post-transcriptional functional involvements of the identified gene markers and provide novel biomarkers for worker surveillance and targets for therapeutic interventions of cardiovascular disease or lung damage [45,46]. Schulte et al. [47] suggested that biomarkers of early responses are necessary for an optimal risk surveillance program for workers in the nanotechnology field.

This study will address this research gap using a systematic, integrated approach combining in vivo, metabolomic, lipidomic, and transcriptomic studies, to evaluate both pulmonary and cardiovascular effects of PEPs exposure. Metabolomic profiles were sequentially assessed following exposure to PEPs to reflect dynamic responses to any pathophysiological changes that may have occurred. Such an approach provides a fresh opportunity for biomarker discovery to elucidate systemic changes occurring following exposure to PEPs. Genome-scale profiles were performed in rat lung tissue and blood to identify important gene signatures relevant to PEPs-induced adverse effects and potential mechanistic pathways. The highly novel integrated in vivo and computational transcriptomic/metabolomic analyses in this study will identify biomarkers of potential PEPs-induced adverse cardiopulmonary effects for occupational/commercial risk surveillance, and reveal important insights into intervention strategies.

## 2. Results

### 2.1. Real-Time Animal Exposure Characterization

The animal exposure characterization data were presented in great detail in a recent publication by the authors [48]. Appendix A summarized the real-time measurements of the mean particle diameter (nm), particle number concentration (10^5^ #/cm^3^), particle mass concentration (µg/m^3^), and count median diameter (nm) of PEPs from laser printer B1 (Appendix A and Appendix A).

The total number concentration during all exposure sessions was approximately 4 × 10^5^–5 × 10^5^ #/cm^3^ with size distributions relatively constant across the 21-day exposure duration [48]. It is also worth noting that as shown in previous studies by the authors, PEPs are a complex mixture of primarily organic and elemental carbon, 1–3% metals (Cu, Ce, Cr, Ni, Fe, and Ti), and low and high molecular weight PAHs [19,21]. Furthermore, tVOCs analysis showed VOCs to be present at relatively low concentrations, with daily averages between 244.8 ± 164.2 parts per billion (ppb) and 363.2 ± 161.7 ppb. The tVOC levels for this study were not measured but expected to be the same as in our previous studies.

### 2.2. mRNA/miRNA Profiling in PEPs-Exposed Rat Lung Tissues

There were 11,010 genes with measured expression in rat lung tissues. At each day of PEPs exposure, genes with a differential expression of at least 1.5-fold change were identified. In a systematic analysis of these genes, the top 5 most significantly (*p* < 0.05) enriched pathways across PEPs exposure time points were listed in Table 1. The top 5 most significantly (*p* < 0.05) impacted biological processes, molecular functions and diseases were included in Appendix A. Top significant (*p* < 0.05) miRNA down-regulators (otherwise notified) of these differentially expressed genes were listed in Table 2.

At day 1 of PEPs exposure, 190 differentially expressed genes were identified in rat lung tissues compared with the control group exposed to HEPA-filtered air. There was perturbation in metabolism, circadian regulation, redox reactions, and immune response, with disease implications in sleep disorder, chemical carcinogenesis, and dysfunctional blood clotting in rat lung (Table 1 and Appendix A). There was no significant miRNA regulator of differentially expressed genes at 1 day of PEPs inhalation (Table 2). It is noteworthy that numerous genes, including *Cyp2c6v1*, *Cyp2c22*, *Cyp3a18*, *Cyp2b3*, *Cyp2c13*, *Ugt2b35*, and *Adh7* in the chemical carcinogenesis pathway, were significantly (*p* < 0.05) changed from their normal expression levels (Table 1) in the PEPs-exposed rat lung on day 1, implying potential tumorigenesis due to laser printer emission exposure.

At day 5 of continuous PEPs inhalation, there were 112 differentially expressed genes in rat lung tissues. Four miRNAs (rno-let-7e-5p, rno-miR-34c-5p, rno-miR-351-5p, and rno-miR-207) were identified as significant (*p* < 0.05) down-regulators of all 112 genes differentially expressed in PEPs-exposed rat lung tissues (Table 2). Exposure induced transcriptional perturbations in rat lung associated with adverse cardiovascular and immune responses, such as viral myocarditis and antigen processing, Type I diabetes mellitus, negative regulation of ATP activity, mental retardation, muscle movements, and skeletal defects (Table 1 and Appendix A).

Of 100 genes differentially expressed in the rat lung at day 9 of PEPs exposure, 3 miRNAs (rno-miR-15b-5p, rno-miR-322-5p, and rno-miR-322-3p) significantly (*p* < 0.05) down-regulated the expression of genes listed in Table 2. PEPs exposure, on Day 9, affected stem cell division, immune response, viral carcinogenesis, and metabolism, with implications in cardiovascular disease and neural disorders manifested by transcriptional perturbations in the PEPs-exposed rat lung (Table 1 and Appendix A).

At day 21 of continuous PEPs exposure, 369 genes were differentially expressed in the rat lung. Among 38 significant (*p* < 0.05) miRNA regulators of these differentially expressed genes, the top 5 miRNAs were rno-miR-135a-5p, rno-miR-29c-3p, rno-miR-143-3p, rno-miR-151-3p, and rno-miR-151-5p (Table 2). Continuous PEPs inhalation at day 21 dysregulated ATP activity and metabolic pathways in rat lung, and was associated with Huntington’s disease, cardiovascular disease, and an autosomal recessive disorder, primary ciliary dyskinesia (Table 1 and Appendix A).

In summary, continuous whole-body inhalation of PEPs affected biological processes and molecular functions of the rat lung related to cardiovascular disease, immune response, metabolism, type I diabetes mellitus, and neural disorders from days 1–21. In particular, cardiovascular dysfunction, metabolic syndrome, and neural disorders were significantly higher in PEPs-exposed rats at every time point (day 1, 5, 9, and 21). These results are concordant with the cardiovascular pathological data (manuscript in review [37]) and toxicology data [48] collected from the same animal study.

### 2.3. mRNA/miRNA Profiling in PEPs-Exposed Rat Blood

There were 14,791 genes with measured expression in rat blood. Differentially expressed genes with a fold change of at least 1.5 were first identified for each time point. Base on the analysis of these differentially expressed genes, the top 5 most significantly (*p* < 0.05) enriched pathways were included in Table 3. The top 5 most affected (*p* < 0.05) biological processes, molecular functions, and diseases were listed in Appendix A. The top significant (*p* < 0.05) miRNA down-regulators of these differentially expressed genes in rat blood were listed in Table 4 for each PEPs-exposure time point.

There were 477 differentially expressed genes identified in rat blood at day 1 of PEPs exposure. These differentially expressed genes were significantly (*p* < 0.05) associated with the integrity of blood vessel wall (including vascular vessel), cellular adhesion, migration, differentiation, proliferation, and apoptosis, hematopoietic stem cell lineage, neuromuscular function, and metabolism. The implicated diseases (*p* < 0.05) included pneumophila, abnormal bleeding and adverse cardiovascular responses (Table 3 and Appendix A). There was no significant (*p* < 0.05) miRNA regulator of these differentially expressed genes (Table 4).

Of 384 genes differentially expressed in rat blood at day 5 of PEPs inhalation, significant (*p* < 0.05) miRNAs regulating these differentially expressed genes were rno-miR-349, rno-miR-200a-3p, rno-miR-129-5p, and rno-miR-31a-5p (Table 4). In brief, 5 days of continuous inhalation of PEPs had a manifested impact on metabolism, inflammatory responses, and various neural disorders in rat blood (Table 3 and Appendix A). It is noteworthy that tumor necrosis binding pathway was dysregulated with differentially expressed genes *Tnfsf13*, *Erap1*, *Tnfsf10*, and *Nucb2* (*p* < 0.05; Appendix A). MiR-200a-3p, essential in epithelial-mesenchymal transition and metastasis [49], was a significant (*p* < 0.05) regulator of numerous differentially expressed genes in rat blood (Table 4). Meningioma, a tumor that arises from the membranes surrounding brain and spinal cord, was among a top significant disease with differentially expressed genes *Klf4* and *Smarce1l* (Appendix A) associated with PEPs inhalation in rat blood. These results indicate considerable PEPs-induced tumorigenic transcriptional dysregulation, and are consistent with the activated pathways of chemical carcinogenesis at day 1 and viral carcinogenesis at day 9 in rat lung (Table 1).

At day 9 of continuous PEPs inhalation, 358 genes were differentially expressed in rat blood. Perturbed regulation of these genes in rat blood affected metabolism and inflammatory responses, and had implications in genetic defects of neural disorders and mental retardation, autoimmune diseases, and viral carcinogenesis (*p* < 0.05; Table 3 and Appendix A). Prominently, viral carcinogenesis pathway was activated in both rat lung and blood at day 9 of PEPs exposure (Table 1; Table 3). The significant (*p* < 0.05) miRNA regulating these differentially expressed genes was rno-miR-365-3p (Table 4).

At day 13 of PEPs exposure, 291 genes were differentially expressed in rat blood. The alteration of these genes was associated with RNA regulation, ATP activities, metabolism, cardiovascular and inflammatory responses, with disease implications in transient neonatal diabetes mellitus and multiple congenital disorders and defects, including postaxial polydactyly, non-syndromic X-linked mental retardation, spondyloepiphyseal dysplasia with congenital joint dislocations, and hereditary hypotrichosis with recurrent skin vesicles syndrome (*p* < 0.05; Table 3 and Appendix A). Rno-miR-378a-3p and rno-miR-378a-5p were the significant (*p* < 0.05) regulators of differentially expressed genes in rat blood at day 13 (Table 4).

There were 271 genes differentially expressed in rat blood at day 17 of PEPs exposure. These genes were over-represented in immune responses, redox reactions, metabolism, GDP/GTP activities, and had implications in type I diabetes mellitus, arthritis, physical disability (Hansen’s disease; leprosis), inflammatory lung disease and skin reaction (*p* < 0.05; Table 3 and Appendix A). No significant miRNA regulator was identified for these 271 differentially expressed genes (Table 4).

A total of 98 differentially expressed genes were identified in rat blood at day 21 of PEPs exposure. These genes were enriched in metabolism, immune responses, ATP activities, and were associated with type I diabetes mellitus and multiple autosomal recessive diseases and abnormalities, including fronto-otopalatodigital osteodysplasia, FLNB-related disorders, spondylocarpotarsal synostosis syndrome, limb-girdle muscular dystrophy type 2A, and Bart-Pumphrey syndrome (*p* < 0.05; Table 3 and Appendix A). Rno-miR-872-5p, rno-miR-124-3p, and rno-miR-30e-5p (rno-miR-30a-5p) were identified as significant down-regulators of the differentially expressed genes (Table 4).

### 2.4. Global Untargeted Profiling of Metabolomics and Lipidomics in Rat Serum

Of the predictive metabolites and lipids, those numbered as 59, 63, 131, 54, 124 and 140 can be mapped with greater than 90% isotypes similarity from Human Metabolome Database (HMDB) and Lipid Maps Database that were differentially expressed (*p* < 0.05) between HEPA and PEPs exposed rats at day 1, 5, 9, 13, 17 and 21 (Appendix A) respectively. Two unknown metabolites were significantly elevated in the PEPs exposed group across all 6 time points (Figure 1A,B).

There were significant enrichments of metabolites and lipids involved in glycerophospholipid, linoleic acid, biotin, biosynthesis of unsaturated fatty acids, alpha-linolenic acid, sphingolipid and histidine metabolism across the various PEP exposure days (Table 5 and Appendix A). Collectively, these pathways associated with a range of established inflammatory, cardiovascular and metabolic disease states like glycogenosis type VII, hypertension, dengue fever, gestational diabetes mellitus, Hartnup disease and histidinemia (Table 6 and Appendix A).

### 2.5. Validation and Quantification of Linoleic Acid, Palmitic Acid and Histidine Pathways in Rat Serum

Given the observation that the linoleic acid pathway, which has an important inflammatory role, was enriched at days 5, 17 and 21, we next validated and quantified linoleic acid, arachidonic acid, docosahexanoic acid (DHA) and palmitic acid using commercial standards. Histidine and histamine were also validated and quantified given their direct roles in the sensitization and human allergy response. Linoleic acid gradually increased significantly (*p* < 0.014) at Day 9 of PEPs exposure, after which it dramatically declined by Day 13. At Day 21, both PEPs and HEPA-exposed samples returned back to baseline (normal) levels (Figure 2A). Arachidonic acid, a downstream metabolite of linoleic acid, was similarly elevated at day 9 of PEPs exposure (*p* < 0.0077) and again significantly reduced at day 13 (*p* = 0.022) and day 17 (*p* = 0.0065) before returning to baseline by Day 21 (Figure 2B). DHA shared a similar profile to linoleic and arachidonic acid (Figure 2C). Palmitic acid, a metabolite feeding into the linoleic acid pathway through conversation to oleic acid, although not significant, was observed to be at lower concentrations in PEPs exposed rat serum compared to HEPA at day 1 through 17 before returning to baseline compared to control (Figure 2D). Histidine quantification is significantly lowered in day 5 (*p* = 0.016) and day 17 (*p* = 0.033) of continuous PEPs inhalation (Figure 3A), and although non-significant, a similar trend was observed with histamine (Figure 3B), a downstream product of histidine.

Combined with the transcriptomic profiles of PEPs-induced mRNA expression changes in rat blood and lung, upstream and downstream genes regulating the identified metabolites were depicted in Figure 2; Figure 3. Genes removed from the initial filtering analysis were not included in the figures, indicating no detectable variation across the samples. Heatmaps of expression of genes regulating linoleic acid, arachidonic acid, docosahexanoic acid (DHA) and palmitic acid in both rat lung and blood across PEPs exposure time points were shown in Figure 2E–H. Time-series mRNA expression of genes regulating histidine and histamine was shown in Figure 3C,D, respectively.

In summary, continuous PEPs inhalation dysregulated genes involved in metabolism, inflammatory/immune responses, and neural functions at every time point (day 1, 5, 9, 13, 17, and 21) in rat blood. The mRNA perturbation observed across various metabolic pathways was consistent with our global metabolomics results in which it was observed to have changed significantly across the corresponding time-points particularly for linoleic and arachidonic acid metabolism, each of which has established inflammatory roles. Pathogenesis of adverse cardiovascular response and disease was remarkable at days 1, 13, and 21 in rat blood. Similarly, an association in hypertension, a risk factor for cardiovascular diseases, was observed at day 5, when increases in linoleic acid pathway activity were established. These results confirm those from the rat lung and the cardiovascular pathological data (manuscript in review [37]). There was significant transcriptional aberration associated with autosomal-recessive disorders in rat blood after continuous PEPs inhalation at days 5, 9 and 21. Genes relevant to various congenital disorders and defects were significantly dysregulated in rat blood at days 13 and 21 of PEPs exposure. PEPs-induced carcinogenic transcriptional perturbation was evident in rat blood at days 5 and 9, consistent with the results from rat lung at days 1 and 9. Continuous PEPs inhalation induced dysregulation of genes and pathways of diabetes at days 13, 17, and 21. Dysregulation of metabolites associated with diabetes were similarly observed at day 5. These results in rat blood suggested linkage of PEPs exposure to cardiopulmonary risk, metabolic syndrome, diabetes, neural disorders, congenital defects, auto-recessive disorders, physical deformation, and carcinogenesis.

## 3. Discussion

PEPs generated from nano-enabled toners are one of the most common types of LCPM released from various nano-enabled products in consumer and occupational use scenarios. The toxicological assessment of PEPs is of utmost importance for worker and consumer health protection. PEPs are deemed highly bioactive in in-vitro and in-vivo studies and may impose significant potential adverse health effects, as previously reported by our group and others [15,28,29,30,31,32,33,34,35,36]. This study utilized a highly novel and integrated, genome-wide mRNA and miRNA profiling in rat lung and blood combined with metabolomics and lipidomics profiling in rat serum to assess PEPs-induced cardiopulmonary risk and other disease implications. The identified PEPs-induced dysregulated genes, pathways, biological processes, molecular functions, and miRNA-mediated transcriptional activities provide important insights into the disease mechanisms and are in agreement with in-vitro and in-vivo toxicological studies of PEPs. Furthermore, the identified important mRNA/miRNA, lipids and metabolites may serve as candidate biomarkers for future occupational and medical surveillance studies.

The respiratory system is the primary route of exposure to airborne particulate matters (PM) via inhalation. Epidemiological studies indicated there is a correlation between inhalation of particles from ambient air and an increased incidence of cardiovascular diseases [50,51]. Numerous in vivo and in vitro studies demonstrated that inhalation exposure to PM causes not only pulmonary damages, but induces adverse cardiovascular effects [52,53,54]. Several mechanisms may be involved in the induction of cardiovascular effects upon inhalation exposure of particulates, particularly those less than PM0.1. First, the translocation of particulates from respiratory system into the blood circulatory through penetrating the air-blood barrier [55]. Second, inhalation of particulates induces pulmonary inflammation, which may mediate a systemic oxidative/inflammatory response that leads to cardiovascular effects [56]. Third, inhalation exposure may trigger neurogenic regulated mechanisms to modulate cardiovascular function [57]. Therefore, evaluation of the changes in gene expression of lung and blood following inhalation exposure to particulates in printer emissions may provide a transcriptomic profiling tool to assess cardiovascular toxicity as well as potential toxicity in other tissue organs. The findings from this study will provide guidance for designing future specific disease model-based studies of PEPs-induced toxicity in a particular tissue organ.

Our group reported that PEPs exposure caused no damage in the histological and chemiluminescence analysis of lung and heart tissues in the in vivo animal studies [48]. However, PEPs exposure elevated sympathetic influence, impaired ventricular performance and repolarization, and caused hypertension and arrhythmia, suggesting increased cardiovascular risk [37]. The results of the global transcriptomic profiling in this study confirmed these pathological data. No significant perturbation in molecular pathways or processes linking to pulmonary diseases was identified in rat lung or blood following PEPs inhalation. In contrast, PEPs inhalation exposure dysregulated genes and pathways involved in cardiovascular malfunction at every observed time point in rat lung and several time points in rat blood. At day 1 of PEPs exposure, *Fgb* was up-regulated (*p* < 0.05) in rat lung, with implications in inherited thrombophilia, afibrinogenemia and dysfibrinogenemia (Appendix A). At day 5 of PEPs inhalation, *RT1-Bb*, associated with dilated cardiomyopathy, viral myocarditis, and Type I diabetes mellitus, was down-regulated (*p* < 0.05) in rat lung (Table 1 and Appendix A). At day 9 of PEPs exposure, *Scn5a* was up-regulated (*p* < 0.05) in rat lung, indicating implications in familial idiopathic ventricular fibrillation, dilated cardiomyopathy, progressive cardiac conduction defect, and progressive familial heart block (Appendix A). At day 13 of PEPs inhalation, genes involved in the regulation of vascular endothelial growth factor production, *Noda1*, *Sulf2*, and *Ccr2*, were down-regulated (*p* < 0.05) in rat blood (Appendix A). At day 21 of PEPs inhalation, *Tnnc1* was down-regulated (*p* < 0.05) in rat lung, with involvements in cardiac muscle contraction and tissue morphogenesis, dilated cardiomyopathy, and hypertrophic cardiomyopathy (Table 1 and Appendix A). *Myh6*, associated with left ventricular noncompaction, was down-regulated (*p* < 0.05) in rat lung (Appendix A). These results are concordant with the cardiovascular pathology data (manuscript in review [37]).

PEPs inhalation exposure also dysregulated ATP activities, sodium and manganese metabolism, and redox functions in rat lung and blood with implications in cardiovascular dysfunctions and blood pressure abnormality. In PEPs exposed rat blood, up-regulation of *Atp6v0e1* at days 1 and 13, down-regulation of *Atp5g1* and *Atp6v1f* at day 1, and up-regulation of *Atp6v1g3* at day 13 were evident (*p* < 0.05; Appendix A). Five days of continuous PEPs inhalation up-regulated *Slco3a1* and *LOC685081* and down-regulated *Slco5a1*, which are associated with sodium-independent organic anion transmembrane transporter activity (*p* < 0.05; Appendix A). Perturbed aldosterone-regulated sodium reabsorption with up-regulation of *Kras* was identified in rat blood at day 9 of PEPs inhalation (*p* < 0.05; Table 3). Manganese ion binding dysfunction with up-regulation of *Ppef2* and down-regulation of *Idi1* was identified in rat lung at day 9 of PEPs exposure (*p* < 0.05; Appendix A). These molecular dysfunctions of sodium and manganese absorption are relevant to high blood pressure, and confirm the observed blood pressure variation in PEPs-exposed rats (manuscript in review [37]). Remarkable disrupted redox functions were identified in rat lung and blood following PEPs inhalation. At day 1, altered oxidoreductase activity and arachidonic acid monooxygenase activity with up-regulation of *Cyp2c22*, *Cyp3a18*, *Cyp2b3*, *Cyp2a1*, *Cyp2c6v1*, *Cyp4a3* and *Cyp3a23/3a1* were identified in rat lung (*p* < 0.05; Appendix A). At 17 days of continuous PEPs inhalation, negative regulation of oxidoreductase activity was ranked a top changed biological process in rat blood with up-regulation of *Cav1* and *Hp* (*p* < 0.05; Appendix A).

The whole genome analysis mirrored the global metabolomics profiles following whole-body PEPs inhalation. Global metabolomics illustrated significant (*p* < 0.05) activation of glycerophospholipid metabolism, linoleic acid metabolism, and biosynthesis of unsaturated fatty acids upon PEPs exposure. In transcriptomics analysis, linoleic acid metabolism pathway was among top 5 significant (*p* < 0.05) pathways in rat lung at Day 1 of PEPs inhalation, with up-regulated genes *Cyp2c6v1*, *Cyp2c22*, *and Cyp3a18* (*p* < 0.05; Table 1). The up-regulation of these 3 genes was also involved in dysregulated steroid hormone biosynthesis, epoxygenase P450 pathway, and endopeptidase activity (Appendix A). At Day 5 of PEPs inhalation, biosynthesis of unsaturated fatty acid pathway was a top altered pathway in rat blood, with up-regulation of *Hacd1* and down-regulation of *Acaa1a* (*p* < 0.05; Table 3). In addition, negative regulation of fatty acid metabolic process was evident in rat blood at Day 17, with up-regulation of *Insig2* (*p* < 0.05; Appendix A).

Activation of the innate immune systems via extracellular and/or intracellular receptors, such as Fc, mannose or Toll-like receptors, can lead to the synthesis of inflammatory lipid mediators like eicosanoids [58,59]. This is in agreement with in-vitro PEPs studies by the authors [60]. These lipid mediators are derived from arachidonic acid, an ω-6 polyunsaturated fatty acid that mammals directly incorporate through diet or by synthesis from linoleic acid [61]. Prostanoids, a major class of eicosanoids, modulates the inflammatory and immune responses [59]. Prostaglandin (PG) E2 is by far the best characterized prostanoid from an immunomodulatoty perspective. Although cytokines are key regulators of immunity, eicosanoids, including PGE2, PGD2, leukotriene (LT) B4, including LTC4, also differentially affect the immune cells by modulating cytokine release, cell differentiation, survival, migration, antigen presentation, and apoptosis. By acting on various aspects of immune function and inflammation, such lipid mediators act as key regulators of the crosstalk between innate and adaptive immunity. Upon early exposure to PEPs (Days 1–9), an up-regulation in linoleic acid, arachidonic acid and DHA metabolism was detected, suggestive of an innate immunity response to PEPs. Particle-generated reactive oxygen species (ROS) can further influence signaling pathways leading to activation of the arachidonic acid cascade [62,63,64]. Studies of titanium oxide nanoparticle inhalation shows significantly impairment of the endothelium-dependent vasoreactivity in coronary arterioles due to increases in microvascular ROS [65]. Similarly, in a previous in-vitro PEPs study by the authors, it was shown that small airway epithelial cells exposure induces cellular effects on human microvascular endothelial cells in an alveolar-capillary co-culture model [28]. The addition of prostanoid formation further contributed to this dysfunction. Deregulation in coronary microvascular function may contribute to cardiac events associated with exposure to particles in this size range [65]. Similar reports have also describe the impact on the arachidonic acid pathway by combustion-derived particles and air pollution [62,66].

Arachidonic acid may also be processed through non-enzymatic reactions. The four double bonds of arachidonic acid are readily oxygenated to form bioactive molecules. Therefore, oxidative stress and/or exposure to ROS and reactive nitrogen species (RNS) results in the oxidation of arachidonic acid, leading to generation of isoprostanes and nitroeicosatetraenoic acids [67], which in turn inhibits COX1, and causes platelet aggregation, vasoconstriction, smooth muscle cell proliferation, and cardiomyocyte hypertrophy [67]. Arachidonic acid enters numerous metabolic pathways that interconnect lipid metabolism with immunity and, therefore, has widespread effect.

DHA possesses anti-inflammatory functions [68]. Resolvins, whose functional roles involve promoting restoration of normal cellular function following inflammation after tissue damage, are a metabolic by-product of omega-3 fatty acid [69,70]. High DHA levels associate with linoleic and arachidonic acid, suggesting an association with a resolving inflammatory response following PEPs inhalation. Inflammation peaked at Day 9 and was suppressed from Day 13 to Day 17 followed by high DHA expression which returned to baseline by Day 21 (Figure 2C).

Histidine is the precursor of histamine production. Metabolomic change in both is, therefore, of interest due to their role in allergen-related sensitization and inflammation [71]. Despite this, a significant decrease in histidine levels, but not histamine levels, was observed in our metabolomic profiles, suggesting PEPs may have minimal roles in inciting the allergic response. Prior studies have shown that L-histidine exhibits antioxidant capabilities, such as scavenging free radicals and chelating divalent metal ions, hence its use to manage oxidative stress [72,73,74]. The physiological effects of histidine include inhibition of pro-inflammatory biomarkers [74]. In vivo and in vitro studies illustrate that administration of histidine lowers the levels of pro-inflammatory markers, such as interleukin 8, interleukin 6, C-reactive protein and tumor necrosis factor alpha [75,76,77]. This indicates that histidine does more than merely scavenge ROS but directly influences pro-inflammatory cytokines [75,76,77]. Mechanisms are likely associated with its potential to increase the amino acid pool and the antioxidant status of cells [78]. Histidine levels are low in patients with heart failure and are correlated with B-type natriuretic peptide, left ventricle ejection fraction, left ventricle end-diastolic volume index, inferior vena cava diameter and the ratio of early diastolic velocity of the mitral inflow to mitral annulus, respectively [79]. L-histidine is an effective inhibitor of spontaneous platelet aggregation and platelet TXB2 generation and l-histidine intake reduces spontaneous platelet aggregation [80]. Such change likely contributes to ROS attenuation in tissues and alleviate clinical symptoms while protecting against intestinal, nervous, and cardiovascular damage.

## 4. Materials and Methods

### 4.1. Whole-Body Inhalation Exposure of Animals

The details of animal exposure and PM characterization were described by the authors in a recent study [48]. In summary, Appendix A (adapted from Pirela et al. [48]) outlines the Printer Exposure Generation System (PEGS) used for the animal exposure. The PEPs-exposed group consisted of Sprague Dawley rats housed in individual cages and exposed via whole-body inhalation to the PEPs and gaseous pollutants of laser printer B1 using the PEGS described in detail in a previous publication [48]. The printer was used to generate PEPs by printing a 5%-page coverage monochrome document using standardized settings [15]. The laser printer B1 was selected based on exposure and toxicological profiles reported in our previous studies [15]. In parallel, another group of animals was exposed to high-efficiency particulate air (HEPA)-filtered air. The system was configured for twelve chambers (one animal per chamber), each equipped with a port for aerosol delivery and temperature and humidity measurement, and a gas sampling port.

Animals were placed in their chambers for 1–5 h for 5 days prior to the exposures to ensure they gradually acclimated to the exposure chambers. The following week, animals were exposed to either PEPs or the room HEPA-filtered air 5 h/day for up to 21 consecutive days. The exposure duration of 21 days, as well as the sacrifice time points (Day 1, 5, 9, 13, 17, and 21), were chosen as sub-chronic exposure periods to provide several delivered doses to the lung and identify any signs of toxicity in the cardiovascular or respiratory systems.

### 4.2. Animal Exposure Characterization (PEPs and Gaseous Co-Pollutants)

#### Real-Time Measurements

The particle number concentration, size distribution, temperature, relative humidity and total volatile organic compounds (tVOC) levels were measured in real time in one of the 12 animal inhalation exposure chambers throughout the exposure durations. A scanning mobility particle sizer (SMPS Model 3080, TSI Inc., Shoreview, MN, USA) was also used for measuring particle number concentration and size distribution (ranging from 2.5 to 210 nm) in the chamber. Real-time particle number concentration as a function of size collected by the SMPS was used to estimate the mass size distribution assuming spherical particles and the density of carbon. Real-time tVOC levels were also monitored using a tVOC monitor (Graywolf Sensing Solutions, Shelton, CT, USA). All the realtime instruments were calibrated, and background tests were performed at the beginning of each sampling experiment. No significant variation in the temperature (°C) and relative humidity (%) in the inhalation animal chambers was observed throughout the exposure period [48].

It is worth noting that a comprehensive chemical analysis of PEPs was performed by the authors and described in detail in previous publications [19,21,48]. In summary, PEPs possess a complex mixture of metals (1–33%) and primarily various organic compounds including polycyclic aromatic hydrocarbon (PAHs; 0.0067%) and other compounds such as elemental carbon (1–3%), organic carbon (42–89%). It is also worth noting that in a recent publication by the authors [19], it was documented that there are synergistic interactions of catalytic metallic nanoparticles with gaseous sVOCs co-pollutants which result in the formation of high molecular and carcinogenic PAHs on the surface of the PEPs.

### 4.3. Animals

One hundred thirty-four healthy, male, 9-week-old Sprague-Dawley rats, weighing an average of 300 g, were purchased from Taconic Farms Inc. (Hudson, NY, USA). The rats were allowed to acclimatize for 1 week before the studies were initiated. Baseline data was collected for 1 week before the exposures occurred. The rats were maintained on a 12-h light/dark cycle and food and water were provided *ad libitum*. Upon arrival, animals were randomly assigned a unique identification number, which determines the exposure group for the animal. There were two treatment groups: (a) Control group: high-efficiency particulate air (HEPA) filtered air, and (b) PEPs exposure group. All the animals were treated humanely in accordance with the guiding principles on the Use of Animals in Toxicology and the Animal Care and Use Committee of Harvard University (Ethical permit No. 04623, 2015).

### 4.4. Whole Blood and Lung Tissue Collection

At the completion of the exposure periods, the animals in each group were anaesthetized with an intraperitoneal injection of a lethal dose (200 mg/kg) of pentobarbital sodium (Anthony Products Co., Arcadia, CA, USA) and sacrificed by exsanguination, followed by whole blood and lung tissue collection. Blood was collected in microfuge tubes pre-loaded with RNAlater and inverted 10 times and transferred to −20 °C or −80 °C for storage. The samples were extracted according to the manufacturer’s instructions or transferred to −80 °C for long term storage. The lung tissue was removed, placed in liquid nitrogen, and stored at −80 °C.

### 4.5. RNA Isolation

Total RNA from frozen lung tissue was extracted using the RNeasy kit, with simultaneous on-column digestion of DNA during RNA purification according to the manufacturer’s instructions (Qiagen, Hilden, Germany). Total RNA isolation from whole blood (stored in RNAlater) was carried out using RiboPure-Blood kit, followed by DNA removal treatment as described by the manufacturer (LifeTechnologies, Carlsbad, CA, USA). Concentration and purity of RNA were quantified by the A 260 nm/A 280 nm and A 260 nm/A 230 nm ratios using a NanoDrop ND-100 spectrophotometer (Thermo Scientific lnc., Waltham, MA, USA). Samples were stored at −80 °C until shipment to the University of Michigan for further processing and analysis.

### 4.6. mRNA and miRNA Microarray Profiling

Global mRNA expression profiles were generated with Rat Gene ST 2.1 plates at the University of Michigan Microarray Facility using an Affymetrix Plus kit (Thermo Fisher Scientific Inc., Waltham, MA, USA). Global miRNA expression profiles were generated with Rat GeneChip miRNA4.1 plates with the same total RNA samples using a FlashTag kit. cDNA was synthesized from 500 ng total RNA, amplified, fragmented, and biotinylated using a GeneChip WT PLUS Reagent kit (Thermo Fisher Scientific Inc., Waltham, MA, USA) according to manufacturer’s instructions. cDNA was then prepared for hybridization with reagents from the Affymetrix GeneTitan Hybridization, Wash, and Stain Kit for WT Array Plates. For hybridization, 2.76 µg was hybridized to the Affymetrix Rat Gene ST 2.1 Arrays, which were then washed, stained, and scanned using the GeneTitan Multi-Channel Instrument according to Affymetrix’s User Guide for Expression Array Plates (P/N 702933 Rev. 2, 2013). Data were analyzed using the Limma, Oligo, and Affy Bioconductor packages implemented in the open source R statistical environment. The robust multi-array average was used to normalize the data and fit log_2_-transformed expression values. The raw microarray data were processed using a robust multi-array average (RMA) method, and expression values were log_2_ transformed, with a principal component analysis used as the final quality control step to visualize mRNA expression values. The analysis was performed with the *oligo* package of Bioconductor in the R statistical environment. The mRNA and miRNA expression profiles in rat lung and blood are available at the National Center for Biotechnology Information (NCBI) Gene Expression Omnibus (GEO) with accession number GSE127951 (with Sub-Series GSE127945, GSE127947, GSE127949, and GSE127950).

### 4.7. mRNA and miRNA Data Analysis

Affymetrix *Transcriptome Analysis Console* (TAC) software was used in miRNA and mRNA data analysis. The microarray data were analyzed with weighted linear models and Bayes methods designed specifically for microarray analysis [81]. Samples were also weighted based on a gene-by-gene update algorithm designed to downweight chips that are deemed less reproducible [82]. Probe sets that had a variance less than 0.05 were removed from further analysis. For mRNA data, only the probe sets listed as “main” by Affymetrix were included for further pathway analysis, which excluded internal controls and un-annotated genes. Probe sets with a fold change of 1.5 or greater were selected. *p*-Values were adjusted for multiple comparisons using the false discovery rate (FDR) with Benjamini and Hochberg tests.

### 4.8. Pathway, Gene Ontology, and Disease Analysis

Significant differentially expressed genes in PEPs exposed vs. HEPA-filtered air exposed rat lung and blood were analyzed with iPathwayGuide (Adviata; Plymouth, MI, USA). These genes were obtained using a threshold of 1 for statistical significance (*p*-value) and a log fold change of expression with the absolute value of at least 0.58. The significance is represented in terms of the negative log (base 10) of the *p*-value. These data were analyzed in the context of pathways obtained from the Kyoto Encyclopedia of Genes and Genomes (KEGG) database (Release 84.0 + /10–26 October 17) [83], gene ontologies (GO) from the Gene Ontology Consortium database (6 November 2017) [84], miRNAs from the miRBase (Release 21) and MICROCOSM (MicroCosm Targets Version 5) databases [85], and diseases from the KEGG database (Release 84.0 + /10–26 October 17) [83]. The top 5 pathways were ranked in terms of the two types of evidence computed by iPathwayGuide: over-representation on the x-axis (pORA) and the total pathway accumulation on the y-axis (pAcc). The two types of evidence, pORA and pAcc, were combined into one final pathway score by calculating a *p*-value using Fisher’s method. This *p*-value was then corrected for multiple comparisons using FDR or Bonferroni corrections.

### 4.9. Prediction of Active miRNAs

The prediction of active miRNAs [86] is based on the enrichment of differentially downregulated target genes of the miRNAs. iPathwayGuide calculates the probability of observing a greater number of differentially downregulated target genes for a given miRNA just by chance. This *p*-value was computed using the hypergeometric distribution.

### 4.10. Lipid and Metabolite Profiling of Rat Serum

#### 4.10.1. Sample Processing

Plasma (100 µL) and 400 µL of lipid/internal standard in isopropanol or acetonitrile were mixed and incubated at 4 °C, with constant agitation at 1400 rpm for 2 h following centrifugation at 4 °C at 4500 rpm for 10 min. Supernatants were then collected for analysis [87].

#### 4.10.2. Metabolite Quantification

One hundred microliters of serum were mixed with 200 µL of methanol-water (50:50 *v*/*v*) by vortexing for 30 s. Following this, the mixture was incubated at 4 °C for 1 h and added to 200 µL chloroform. Following vigorously vortexing (30 s) and centrifugation (14,000 rpm for 10 min at 4 °C), proteins were precipitated. The organic and aqueous phases were next collected separately and dried using vacuum centrifugation. The aqueous phase was then reconstituted in 200 µL of mobile phase E:F (25:75 *v*/*v*, mobile phase E-30 mM ammonium formate in water and mobile phase F-30 mM ammonium formate in 85:15 ACN: Water) and used for histamine and histidine analysis. The organic phase was reconstituted in 100 µL ethanol for fatty acid analysis.

#### 4.10.3. Histamine and Histidine Analysis

LC-MS/MS histamine and histidine analysis was performed using an ACQUITY UPLC system (Waters, Milford, MA, USA) coupled to a Xevo TQ-MS system (Waters, Milford, MA, USA). Sample (10 µL) were separated using A Waters ACQUITY BEH Amide (1.7 µM; 2.1 × 150 mM) at 40 °C in ESI Positive mode for the aqueous phase containing histamine and histidine with eleution gradient using mobile phases C and D (flow rate 0.4 mL/min) at 75% D linearly decreasing to 45% for 2 min and held for 1 min before equilibrating the system for the next injection. For mass spectrometry, we used a capillary voltage of 3.4 kV, desolvation temperature 400 °C, cone gas flow 150 L/h, desolvation gas flow 800 L/h and a source temperature of 150 °C for analysis. Corresponding parameters for histamine and histidine analysis were 2.5 kV, 450 °C, 0 L/h, 900 L/h, and 150 °C, respectively. MassLynx and TargetLynx were used for data acquisition quantification respectively. Statistical analysis was carried out using 2-way ANOVA followed by post-hoc multiple correction using FDR [88]. Details of lipid and metabolite profiling were included in Appendix A.

## 5. Conclusions

The current study is in agreement with other in vivo and in vitro studies, linking exposure to nanoparticles to transcriptomic alterations associated with cardiovascular and inflammatory diseases [5,24,30,35,60,89,90]. In addition to cardiovascular disease and metabolic syndrome, continuous PEPs inhalation induced gene expression changes associated with neural disorders in both rat lung and blood at every observed time point. Our group plan to validate the identified genes and metabolites in rat heart tissues as well as blood samples from occupational workers in our ongoing and future studies. Furthermore, the systematic analysis suggested PEPs exposure induced transcriptional aberrations associated with other diseases in rats, including diabetes, congenital defects, auto-recessive disorders, physical deformation, and carcinogenesis. These findings provide guidance for designing future in vivo studies using a disease-specific animal model for assessing PEPs exposure-induced toxicity in a particular tissue organ. This study provides evidence for future biomarker-based surveillance and mechanistic studies of PEPs-induced diseases. This study is novel and to our knowledge, the first to integrate in vivo, transcriptomics, metabolomics, and lipidomics to assess PEPs inhalation exposure-induced disease risks using a rat model.

## Figures and Tables

**Figure 1 ijms-20-06348-f001:**
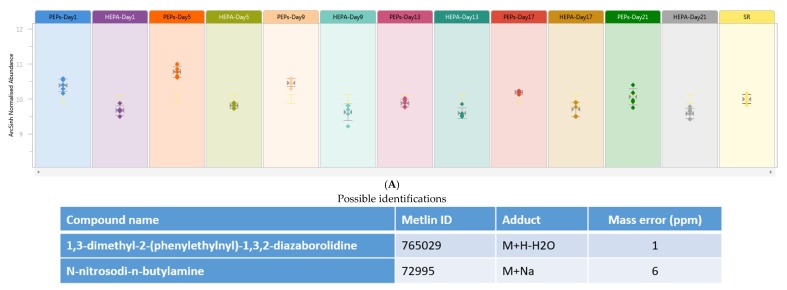
Normalized abundance of compounds 1 and 2 across the exposure days between PEPs exposed and HEPA control rats. Both compounds 1 and 2 are significantly (*p* < 0.05) elevated in the serum of PEPs exposed rats across the six time points. (**A**) Compound 1 (0.75_181.1299 *m*/*z*) abundance; (**B**) Compound 2 (2.63_443.2326 *m*/*z*) abundance.

**Figure 2 ijms-20-06348-f002:**
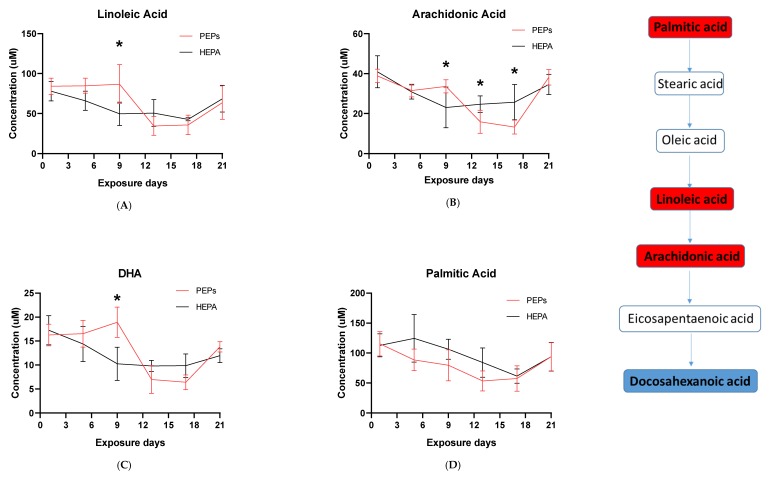
Quantification of linoleic acid, arachidonic acid, docosahexanoic acid (DHA) and palmitic acid in PEPs and HEPA control inhalation rat serums and heatmaps of genes regulating these metabolites in rat blood and lung. Fatty acids that were quantified (bold) are illustrated in the pathway. Palmitic acid, linoleic acid, arachidonic acid plays a role in pro-inflammatory response (red box) while anti-inflammatory are shown in blue box. (**A**) Linoleic acid, (**B**) Arachidonic acid, (**C**) Docosahexanoic acid and (**D**) Palmitic acid. *—Adjusted *p*-value < 0.05. Heatmaps depict log fold change (LogFC) of genes regulating Linoleic acid, Arachidonic acid, Docosahexaenoic acid (DHA), and Palmitic acid in both rat blood and lung samples at each day. (**E**) Genes regulating linoleic Acid, (**F**) Genes regulating arachidonic acid, (**G**) Genes regulating docosahexaenoic acid, (**H**) Genes regulating palmitic acid in rat blood and lung samples. *—0.01 < *p*-value < 0.05, **—*p*-value < 0.01.

**Figure 3 ijms-20-06348-f003:**
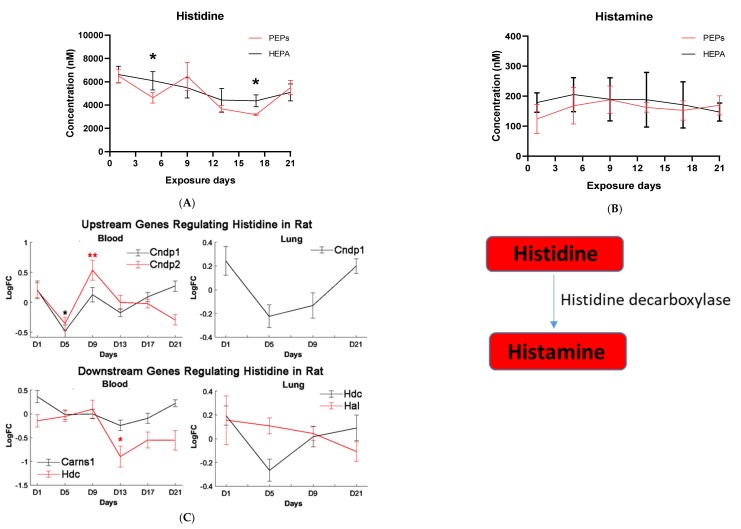
Quantification of histidine and histamine in PEPs and HEPA control inhalation rat serums and expression of genes regulating histidine and histamine in both rat blood and lung samples. Metabolites that were quantified (bold) are illustrated in the pathway. Histidine and histamine plays a role in pro-inflammatory response (**A**) Histidine and (**B**) Histamine. *—Adjusted *p*-value < 0.05. Log fold change (LogFC) of mRNA expression of genes regulating Histidine and Histamine in both blood and lung samples at each day. (**C**) Genes regulating Histidine both rat blood and lung samples, (**D**) Genes regulating Histamine both rat blood and lung samples. *—0.01 < *p*-value < 0.05, **—*p*-value < 0.01. Genes removed from the initial filtering analysis were not included in the figure, indicating no detectable variation across the samples.

**Table 1 ijms-20-06348-t001:** Top 5 significant (*p* < 0.05) KEGG pathways in PEPs-exposed rat lung tissues. The listed genes had a fold change of at least 1.5 in rat lung tissues exposed to PEPs vs. HEPA-filtered air control group. Genes with a statistically significant (*p* < 0.05) differential expression are highlighted in bold.

Exposure Time Point	KEGG Pathway	Up-Regulated Genes in the Pathway	Down-Regulated Genes in the Pathway
Day 1	Retinol Metabolism	***Cyp2c22, Cyp3a18, Cyp2b3, Cyp4a3, Ugt2b35, Cyp2a1, Cyp2c6v1,*** *Cyp1a1, Cyp2a3, Cyp2c11, Cyp2c13, Cyp3a23/3a1*	***Adh7, Dhrs9***
	Chemical Carcinogenesis	***Cyp2c6v1, Cyp2c22, Cyp3a18, Cyp2b3, Ugt2b35*** *, Cyp1a1, Gsta2, Cyp2c11, Cyp2c13, Cyp3a23/3a1*	***Adh7***
	Steroid Hormone Biosynthesis	***Cyp2c6v1, Cyp2c22, Cyp3a18, Cyp2b3, Cyp2c13, Ugt2b35,*** *Cyp1a1, Cyp2c11, Sult1e1, Cyp3a23/3a1*	
	Linoleic Acid Metabolism	***Cyp2c6v1, Cyp2c22, Cyp3a18,*** *Cyp2c11, Cyp2c13, Cyp3a23/3a1*	
	Circadian Rhythm	***Nr1d1, Per2, Per3, Bh1he40***	***Arntl***
Day 5	Viral Myocarditis		***RT1-Bb, Cycs,*** *RT1-CE16, RT1-N1*
	Herpes Simplex Infection		***Pilra, RT1-Bb, Cycs,*** *RT1-CE16, Ifit1, LOC100910669, RT1-N1*
	Antigen Processing and Presentation	*Klrc3*	***RT1-Bb,*** *RT1-CE16, RT1-N1*
	Cardiac Muscle Contraction		*Tnnc1, Tnni3, Tnnt2, Cox6a2*
	Type I Diabetes Mellitus		***RT1-Bb,*** *RT1-N1, RT1-CE16*
Day 9	Terpenoid Backbone Biosynthesis		***Hmgcs1*** *, **Idi1***
	Herpes Simplex Infection	*Ifit1, C5, RT1-CE16, Per2*	
	Dilated Cardiomyopathy		*Tnnc1, RGD1565478*
	Viral Carcinogenesis	***Hist1h4m,*** *RT1-CE16*	
	Systemic Lupus Erythematosus	***Hist1h4m,*** *C5*	*RGD1565478*
Day 21	Huntington’s Disease		***Dnah5*** *, **Dnai2**, **Dnai1**, **Dnah12**, **Dnah6**, **Dnali1**, **Dnah7**, **Dnah3**, **Dnah11**, **Dnah14,** Cox6a2, Dnah9*
	Hypertrophic Cardiomyopathy		***Tnnc1*** *, Tnnt2, Myh6, Tnni3, Ryr2, Actc1, Mybpc3, Ttn*
	Circadian Rhythm	***Cry1, Per2*** *, **Rorc,** Rorb,*	***Npas2***
	Cardiac Muscle Contraction		***Tnnc1*** *, Tnnt2, Myh6, Tnni3, Ryr2, Cox6a2, Actc1*
	Dilated Cardiomyopathy		***Tnnc1*** *, Tnnt2, Myh6, Tnni3, Ryr2, Actc1, Mybpc3, Ttn*

**Table 2 ijms-20-06348-t002:** Top identified miRNA targeting multiple differentially expressed genes in rat lung tissues. Predicted miRNAs with a significant *p*-value (*p* < 0.05) in the enrichment analysis are listed. Genes listed in the table were down-regulated by the corresponding miRNA unless otherwise specified with an asterisk (*).

Exposure Time Point	miRNA	Differentially Expressed Target Genes
Day 1	−	*−*
Day 5	rno-let-7e-5p	*Bpifa1, RGD1359158, RT1-Bb, Fcrl2, Myom2*
	rno-miR-34c-5p	*C6, Cd177, Akr1c12, Csrp3*
	rno-miR-351-5p	*RGD1566006, Tnnc1, Ahrr, Cox6a2*
	rno-let-7b-5p	*C6, Mb, Cd177, Myom2*
Day 9	rno-miR-15b-5p	*Gper1, Sln, Stox1, Styk1, LOC102550375*
	rno-miR-322-5p (rno-miR-322-3p)	*Gper1, Olr384, Zfp623, Sln, Styk1*
Day 21	rno-miR-135a-5p	*RGD1359158, Scgb3a1, Enpp3, Pon1, Tspan1, Cerkl, Efhd1, Cfap45, Spa17, Aoc1, Ffar4, Ccdc153, Wfdc21, Olr404, Fv1, Ces2g, Rgs16**
	rno-miR-29c-3p	*Fabp3, Olr297, Rsph4a, Eln, Tspan1, Spa17, Tsnaxip1, Kif24, Ffar4, Npepo, Gpr37*
	rno-miR-143-3p	*Bpifa1, Chst9, Pon1, Rarres1, Lrrc23, Bpifa5, Ribc2, RGD1304810, Saxo2, Efhb, Dnali1, Npepo, Dnah11*
	rno-miR-151-3p (rno-miR-151-5p)	*Tnnc1, Sln, Fabp3, Srl, Tmc5, Rtn4ip1, Cfap206, Ttc21a, Saxo2, Hrc, Foxa3, Riiad1, Cd8a* *

* Up-regulated genes.

**Table 3 ijms-20-06348-t003:** Top 5 significant (*p* < 0.05) KEGG pathways in PEPs-exposed blood. The listed genes had a fold change of at least 1.5 in rat blood exposed to PEPs vs. HEPA-filtered air control group. Genes with a statistically significant (*p* < 0.05) differential expression are highlighted in bold.

Exposure Time Point	KEGG Pathway	Up-Regulated Genes in the Pathway	Down-Regulated Genes in the Pathway
Day 1	Platelet Activation	***Gp5*, *Gp1bb*, *Gp9*, *Gp1ba*, *Itga2b*, *Vwf*, *P2rx1*, *P2ry12*, *Itgb3*, *Gucy1a3*, *Ptgs1***	***F2r*, *Mapk13***
	ECM-Receptor Interaction	***Col6a5*, *Gp5*, *Gp1bb*, *Gp9*, *Gp1ba*, *Sv2a*, *Itga2b*, *Vwf*, *Itgb3***, *Itga6*	
	Hematopoietic Cell Lineage	***Gp5*, *Gp1bb*, *Gp9*, *Gp1ba*, *Itga2b*, *Cd9*, *Itgb3***	
	Legionellosis	*Naip6*, *Naip5*	*Casp9*, *Cycs*
	Olfactory Transduction	***Olr220*, *Olr259*, *Olr712*, *Olr352*, *Olr120*, *Olr1206*, *Olr1073*, *Olr472*, *Olr756*, *Olr1433*, *Olr613*, *Olr920*, *Olr1143*, *Olr126*, *Olr1093*, *Olr1l*, *Olr60*, *Olr1274*, *LOC500460***, *Olr214*, *Olr67*, *Olr816*, *Olr1385*, *Olr1229*, *Olr298*, *Olr380*	***Olr777*, *Olr906*, *Olr103***, *Olr204*, *Olr168*, *Olr300*
Day 5	Olfactory Transduction	***Olr1422*, *Olr748*, *Olr480*, *Olr130*, *Olr252*, *Olr1369*, *Olr448*, *Olr611*, *Olr1401*, *Olr569*, *Olr199*, *Olr1471*, *Olr140*, *Olr1230*, *Olr808*, *Olr119*, *Olr826*, *Olr499*, *Olr230*, *Olr168***, *Olr1539*, *Olr1192*, *Olr1332*, *Olr255*, *Olr1619*	***Gng13*, *Olr240*, *Olr339***, *Olr1460*, *Olr1196*, *Olr1736*, *Olr67*, *Olr1537*, *Olr1569*
	Chemokine Signaling Pathway	*Ppbp*	***Plcb1*, *Gngt2*, *Xcr1*, *Gng13*, *Ccr1*, *Cx3cr1***, *Ccr1l1*
	Biosynthesis of Unsaturated Fatty Acid	***Hacd1***	***Acaa1a***, *Acot3*
	Protein Processing in Endoplasmic Reticulum	*Prkn*	***Wfs1***, *Ppp1r15a*
	Complement and Coagulation Cascades	***Pros1***	
Day 9	Alcoholism	***Kras*, *Gng10*, *Hist1h2bd*, *LOC100910200*, *Slc18a2*,** ***H3f3b***	***Hist2h2aa3*, *Hist1h2aa*, *Gng13***,***LOC685069***, *Hist1h2bf*, *Hist2h2ac*, *Pkia*
	Systemic Lupus Erythematosus	***H3f3b*, *Hist1h2bd*, *LOC100910200***, *RT1-Bb*	***Hist2h2aa3*, *His1h2aa***, ***LOC685069*****,***Hist1h2bf*, *Hist2h2ac*
	Aldosterone-Regulated Sodium Reabsorption	***Kras*,***LOC298795*, *Ins2*, *Fxyd2*	
	Glycosylphosphatidylinositol (GPI)-Anchor Biosynthesis	***Pigw***, *Pigo*	***Pigz***
	Viral Carcinogenesis	***Kras*, *Ccr5*, *Hist1h2bd*, *LOC100910200*, *RT1-CE16***	***Cdc20*,***Hist1h2bf*, *RT1-M6-1*, *Skp2*
Day 13	Olfactory Transduction	***Olr1070*,** *Olr395*	***Olr1695*, *Olr951*, *Olr837*, *LOC103693627*, *Olr399*, *Olr1405*, *Olr1061*, *Olr1662*, *Olr1499*, *Olr1058*, *Olr956*, *Olr1376*, *Olr1353***, *Olr122*, *Olr168*, *Olr1619*, *Olr1002*, *Olr300*
	Ribosome	***Rpl30*, *Rsl24d1*, *Rpl15*, *Rpl18*,** *Rpl37a*	
	Collecting Duct Acid Secretion	***Atp6v0e1*, *Atp6v1g3***	
	Histidine Metabolism	***Hdc***, *Maob*	
	mRNA Surveillance Pathway	***Magoh***, *Sap18*	
Day 17	Type I Diabetes Mellitus	***RT1-CE11***, *Ins2*, *RT1-CE16*	***RT1-DOa*, *RT1-T24-1*, *RT1-M10-1***, *RT1-CE5*, *RT1-T24-3*, *RT1-CE15*
	Antigen Processing and Presentation	***RT1-CE11***,***Klrc3***, *RT1-CE16*	***RT1-DOa*, *RT1-T24-1*, *RT1-M10-1***, *RT1-CE5*, *RT1-T24-3*, *RT1-CE15*
	Graft-Versus-Host Disease	***RT1-CE11***, *RT1-CE16*	***RT1-DOa*, *RT1-T24-1*, *RT1-M10-1***, *RT1-CE5*, *RT1-T24-3*, *RT1-CE15*
	Allograft Rejection	***RT1-CE11***, *RT1-CE16*	***RT1-DOa*, *RT1-T24-1*, *RT1-M10-1***, *RT1-CE5*, *RT1-T24-3*, *RT1-CE15*
	Cell Adhesion Molecules (CAMs)	***RT1-CE11***,***Selp***, *Nrcam*, *RT1-CE16*	***RT1-DOa*, *Madcam1*, *RT1-T24-1*, *RT1-M10-1***, *RT1-CE5*, *RT1-T24-3*, *RT1-CE15*
Day 21	Arachidonic Acid Metabolism		***Alox12e*, *Ptges3l1***, *Hpgds*
	Type I Diabetes Mellitus	*Cpe*	***Gzmb***
	Ferroptosis		***Hmox1***
	Mineral Absorption		***Hmox1***
	Vitamin B6 Metabolism	*RGD1566085*	

**Table 4 ijms-20-06348-t004:** Top identified miRNA targeting multiple differentially expressed genes in PEPs-exposed rat blood. Predicted miRNAs with a significant *p*-value (*p* < 0.05) in the enrichment analysis are listed. Genes listed in the table were down-regulated by the corresponding miRNA unless otherwise specified with an asterisk (*).

Exposure Time Point	miRNA	Differentially Expressed Target Genes
Day 1	−	−
Day 5	rno-miR-349	*Try10, Ppm1h, Nags, Kif13b, Atg101, P2ry2, Slc16a3, Trmt44*
	rno-miR-200a-3p	*Ca5b, Pi16, Krtap3-3, Rpl10l, Olr1196, Tmem220, Slc16a3, Cx3cr1, Ndufa2 **
	rno-miR-129-5p	*Adam6, Sulf2, Slc26a6, Thtpa, LOC500124, Tnfsf10, Mrgprf, Gria3 **
	rno-miR-31a-5p	*Ca5b, Supt4h1, Rpl13, Kif13b, Ell3, Doc2g, Cdca3, LOC500124, Tp53rk, Sec11c *, Olr748 **
Day 9	rno-miR-365-3p	*Pclaf, Olr721, Rpl13, Cd160, Glod5, Olr713*
Day 13	rno-miR-378a-3p (rno-miR-378a-5p)	*Fbln1, Doc2a, Andpro, LOC102546864, Hist2h2ab, Tmem222, Oas1a, Atp6v0e1 **
Day 17	−	*−*
Day 21	rno-miR-872-5p	*Plekhb1, Gzmb, Trappc2l, Npl, Lrrc61, Hmox1, Hpgds*
	rno-miR-124-3p	*RGD1359158, Clic2, Trappc2l, Rtn4ip1, Supt4h1*
	rno-miR-30e-5p (rno-miR-30a-5p)	*Fv1, Ces1c, Alox12e, Ldhb, Gzmb*

* Up-regulated genes.

**Table 5 ijms-20-06348-t005:** Top 5 associated pathways and their respective compounds across the different exposure days in pathway analysis. Pathways that are significantly different (*p* < 0.05) between PEPs-exposed and HEPA control rat serum are highlighted bold. Compounds that are significantly different (*p* < 0.05) in PEPs-exposed rat serum are in red text.

	Pathway Analysis	Compound
Day 1	Glycerophospholipid Metabolism	Phosphatidylethanolamine; **Phosphatidylcholine**; Dihydroxyacetone phosphate; **LysoPC(18:1(9Z))**; 1,2-Diacyl-sn-glycerol; Citicoline; Phosphorylcholine; Choline; Acetylcholine ; *O*-Phosphoethanolamine; Ethanolamine; PA(16:0/16:0); Acyl-CoA; 1-Acyl-sn-glycerol 3-phosphate; CDP-diacylglycerol; Glycerol 3-phosphate; 1-Acyl-sn-glycero-3-phosphoethanolamine; 2-Acyl-sn-glycero-3-phosphoethanolamine; 2-Acyl-sn-glycero-3-phosphocholine; CDP-glycerol; PS(16:0/16:0); Phosphatidylglycerol; Dihydroxyacetone Phosphate Acyl Ester; CDP-Ethanolamine; 1-Phosphatidyl-d-myo-inositol; Glycerylphosphorylethanolamine; Glycerophosphocholine; Phosphatidyl-*N*-methylethanolamine; Phosphatidylglycerophosphate; Cardiolipin
	Linoleic Acid Metabolism	Linoleic acid; **Phosphatidylcholine**; 9,10-Epoxyoctadecenoic acid; 12,13-EpOME; 13-l-Hydroperoxylinoleic acid
	Biotin Metabolism	Biotin; Biotinyl-5′-AMP; Biocytin; Holo-[carboxylase]; **l-Lysine**
	Pyrimidine Metabolism	Uridine; Uracil
	Alpha-Linolenic Acid Metabolism	Thioredoxin; Uridine 5′-diphosphate; l-Glutamine; Carbamoylphosphate; 4,5-Dihydroorotic acid; Orotidylic acid; RNA; Uridine triphosphate; Uridine 5′-monophosphate; **Uridine**; Dihydrouracil; Ureidopropionic acid; Cytidine triphosphate; CDP; Cytidine monophosphate; Cytidine; Thioredoxin disulfide; dCDP; dCTP; dCMP; Deoxycytidine; Deoxyuridine triphosphate; dUDP; dUMP; Deoxyuridine; Thymidine 5′-triphosphate; dTDP; 5-Thymidylic acid; Thymidine; Dihydrothymine; Ureidoisobutyric acid; P1,P4-Bis(5′-uridyl) tetraphosphate; Ureidosuccinic acid; Phosphoribosyl pyrophosphate; Orotic acid; **Uracil**; Beta-Alanine; DNA; Deoxyribose 1-phosphate; Thymine; 3-Aminoisobutanoic acid
Day 5	**Biosynthesis of Unsaturated Fatty Acids**	(13Z,16Z)-Docosadi-13,16-enoyl-CoA; Tetracosenoyl-CoA; Docosenoyl-CoA; Icosenoyl-CoA; Tetracosanoyl-CoA ; Docosanoyl-CoA; (7Z,10Z,13Z,16Z)-Docosatetraenoyl-CoA; (11Z,14Z)-Icosadienoyl-CoA; (7Z,10Z,13Z,16Z,19Z)-Docosapentaenoyl-CoA; (11Z,14Z,17Z)-Icosatrienoyl-CoA; Palmityl-CoA; Stearoyl-CoA; Eicosanoyl-CoA; Oleoyl-CoA; Linoleoyl-CoA; Arachidonyl-CoA; 8,11,14-Eicosatrienoyl-CoA; Gamma-linolenoyl-CoA; (4Z,7Z,10Z,13Z,16Z,19Z)-Docosahexaenoyl-CoA; (5Z,8Z,11Z,14Z,17Z)-Icosapentaenoyl-CoA; Alpha-Linolenoyl-CoA; 13,16-Docosadienoic acid; Nervonic acid; Erucic acid; Icosenoic acid; Tetracosanoic acid; Behenic acid; **7,10,13,16-Docosatetraenoic acid**; Icosadienoic acid; **Clupanodonic acid**; Icosatrienoic acid; Palmitic acid; Stearic acid; Arachidic acid; **Oleic acid**; **Linoleic acid**; **Arachidonic acid**; 8,11,14-Eicosatrienoic acid; Gamma-Linolenic acid; **(4Z,7Z,10Z,13Z,16Z,19Z)-Docosahexaenoic acid**; Eicosapentaenoic acid; **Alpha-Linolenic acid**
	**Linoleic Acid Metabolism**	**Linoleic acid**; **Phosphatidylcholine**; 9,10-Epoxyoctadecenoic acid; 12,13-EpOME; 13-L-Hydroperoxylinoleic acid
	**Alpha-Linolenic Acid Metabolism**	OPC4-CoA; OPC6-CoA; OPC8-CoA; **Phosphatidylcholine**; **Alpha-Linolenic acid**; trans-2-Enoyl-OPC4-CoA; trans-2-Enoyl-OPC6-CoA; trans-2-Enoyl-OPC8-CoA; Stearidonic acid
	Phenylalanine, Tyrosine and Tryptophan Biosynthesis	Phenylpyruvic acid; **l-Phenylalanine**; l-Tyrosine; 4-Hydroxyphenylpyruvic acid
	Glycerophospholipid Metabolism	Phosphatidylethanolamine; **Phosphatidylcholine**; Dihydroxyacetone phosphate; **LysoPC(18:1(9Z))**; 1,2-Diacyl-sn-glycerol; Citicoline; Phosphorylcholine; Choline; Acetylcholine ; *O*-Phosphoethanolamine; Ethanolamine; PA(16:0/16:0); Acyl-CoA; 1-Acyl-sn-glycerol 3-phosphate; CDP-diacylglycerol; Glycerol 3-phosphate; 1-Acyl-sn-glycero-3-phosphoethanolamine; 2-Acyl-sn-glycero-3-phosphoethanolamine; 2-Acyl-sn-glycero-3-phosphocholine; CDP-glycerol; PS(16:0/16:0); Phosphatidylglycerol; Dihydroxyacetone Phosphate Acyl Ester; CDP-Ethanolamine; 1-Phosphatidyl-d-myo-inositol; Glycerylphosphorylethanolamine; Glycerophosphocholine; Phosphatidyl-*N*-methylethanolamine; Phosphatidylglycerophosphate; Cardiolipin
Day 9	**Biosynthesis of Unsaturated Fatty Acids**	(13Z,16Z)-Docosadi-13,16-enoyl-CoA; Tetracosenoyl-CoA; Docosenoyl-CoA; Icosenoyl-CoA; Tetracosanoyl-CoA ; Docosanoyl-CoA; (7Z,10Z,13Z,16Z)-Docosatetraenoyl-CoA; (11Z,14Z)-Icosadienoyl-CoA; (7Z,10Z,13Z,16Z,19Z)-Docosapentaenoyl-CoA; (11Z,14Z,17Z)-Icosatrienoyl-CoA; Palmityl-CoA; Stearoyl-CoA; Eicosanoyl-CoA; Oleoyl-CoA; Linoleoyl-CoA; Arachidonyl-CoA; 8,11,14-Eicosatrienoyl-CoA; Gamma-linolenoyl-CoA; (4Z,7Z,10Z,13Z,16Z,19Z)-Docosahexaenoyl-CoA; (5Z,8Z,11Z,14Z,17Z)-Icosapentaenoyl-CoA; Alpha-Linolenoyl-CoA; 13,16-Docosadienoic acid; Nervonic acid; Erucic acid; Icosenoic acid; Tetracosanoic acid; Behenic acid; **7,10,13,16-Docosatetraenoic acid**; Icosadienoic acid; Clupanodonic acid; Icosatrienoic acid; **Palmitic acid**; **Stearic acid**; Arachidic acid; Oleic acid; Linoleic acid; Arachidonic acid; 8,11,14-Eicosatrienoic acid; Gamma-Linolenic acid; **(4Z,7Z,10Z,13Z,16Z,19Z)-Docosahexaenoic acid**; Eicosapentaenoic acid; Alpha-Linolenic acid
	**Glycerophospholipid Metabolism**	Phosphatidylethanolamine; **Phosphatidylcholine**; Dihydroxyacetone phosphate; **LysoPC(18:1(9Z))**; 1,2-Diacyl-sn-glycerol; Citicoline; Phosphorylcholine; Choline; Acetylcholine ; *O*-Phosphoethanolamine; Ethanolamine; PA(16:0/16:0); Acyl-CoA; 1-Acyl-sn-glycerol 3-phosphate; CDP-diacylglycerol; Glycerol 3-phosphate; 1-Acyl-sn-glycero-3-phosphoethanolamine; 2-Acyl-sn-glycero-3-phosphoethanolamine; 2-Acyl-sn-glycero-3-phosphocholine; CDP-glycerol; PS(16:0/16:0); Phosphatidylglycerol; Dihydroxyacetone Phosphate Acyl Ester; CDP-Ethanolamine; 1-Phosphatidyl-d-myo-inositol; Glycerylphosphorylethanolamine; **Glycerophosphocholine**; Phosphatidyl-*N*-methylethanolamine; Phosphatidylglycerophosphate; Cardiolipin
	Retinol Metabolism	**Vitamin A**; Retinal; B-Carotene; 11-*cis*-Retinal; Retinyl ester; 9-*cis*-Retinol; 11-*cis*-Retinol; **Retinoic acid**; 9-*cis*-Retinal; Retinyl palmitate; All-trans-13,14-dihydroretinol; 11-*cis*-Retinyl palmitate; all-*trans*-4-Hydroxyretinoic acid; all-*trans*-18-Hydroxyretinoic acid; all-*trans*-5,6-Epoxyretinoic acid; 9-*cis*-Retinoic acid; Retinoyl b-glucuronide
	Tryptophan Metabolism	**l-Tryptophan**; Melatonin; *N*-Acetylserotonin; Serotonin; **5-Hydroxyindoleacetic acid**; 5-Hydroxykynurenamine; 5-Hydroxykynurenine; 5-Hydroxy-l-tryptophan; l-Formylkynurenine; Acetoacetyl-CoA; (S)-3-Hydroxybutanoyl-CoA; Crotonoyl-CoA; Glutaryl-CoA; Oxoadipic acid; 2-Amino-3-carboxymuconic acid semialdehyde; 3-Hydroxyanthranilic acid; l-Kynurenine; Formylanthranilic acid; l-3-Hydroxykynurenine; 3-Hydroxykynurenamine; **Indoleacetaldehyde**; 5-Hydroxy-*N*-formylkynurenine; 5-Hydroxyindoleacetaldehyde; Tryptamine; Indoleacrylic acid; Acetyl-*N*-formyl-5-methoxykynurenamine; 6-Hydroxymelatonin; Formyl-5-hydroxykynurenamine; 5-Methoxyindoleacetate; 4,6-Dihydroxyquinoline; Acetyl-CoA; 2-Aminomuconic acid semialdehyde; 2-Aminobenzoic acid; l-Tryptophanyl-tRNA(Trp); Cinnavalininate; 4-(2-Amino-3-hydroxyphenyl)-2,4-dioxobutanoic acid; 4-(2-Aminophenyl)-2,4-dioxobutanoic acid; 4,8-Dihydroxyquinoline; Indoleacetic acid; *N*-Methylserotonin; *N*-Methyltryptamine
	Linoleic Acid Metabolism	Linoleic acid; **Phosphatidylcholine**; 9,10-Epoxyoctadecenoic acid; 12,13-EpOME; 13-l-Hydroperoxylinoleic acid
Day 13	**Glycerophospholipid Metabolism**	**Phosphatidylethanolamine**; **Phosphatidylcholine**; Dihydroxyacetone phosphate; LysoPC(18:1(9Z)); 1,2-Diacyl-sn-glycerol; Citicoline; Phosphorylcholine; Choline; Acetylcholine ; *O*-Phosphoethanolamine; Ethanolamine; PA(16:0/16:0); Acyl-CoA; 1-Acyl-sn-glycerol 3-phosphate; CDP-diacylglycerol; Glycerol 3-phosphate; 1-Acyl-sn-glycero-3-phosphoethanolamine; 2-Acyl-sn-glycero-3-phosphoethanolamine; 2-Acyl-sn-glycero-3-phosphocholine; CDP-glycerol; **PS(16:0/16:0)**; Phosphatidylglycerol; Dihydroxyacetone Phosphate Acyl Ester; CDP-Ethanolamine; 1-Phosphatidyl-d-myo-inositol; Glycerylphosphorylethanolamine; Glycerophosphocholine; Phosphatidyl-*N*-methylethanolamine; Phosphatidylglycerophosphate; Cardiolipin
	**Linoleic Acid Metabolism**	Linoleic acid; **Phosphatidylcholine**; 9,10-Epoxyoctadecenoic acid; 12,13-EpOME; 13-l-Hydroperoxylinoleic acid
	Alpha-Linolenic Acid Metabolism	OPC4-CoA; OPC6-CoA; OPC8-CoA; **Phosphatidylcholine**; Alpha-Linolenic acid; trans-2-Enoyl-OPC4-CoA; trans-2-Enoyl-OPC6-CoA; trans-2-Enoyl-OPC8-CoA; Stearidonic acid
	Glycosylphosphatidylinositol(GPI)-Anchor Biosynthesis	UDP-N-acetyl-d-glucosamine; 1-Phosphatidyl-d-myo-inositol; (GlcN)1 (Ino(acyl)-P)1; Dolichyl phosphate d-mannose; (GlcN)1 (Ino(acyl)-P)1 (Man)3 (EtN)1 (P)1; **Phosphatidylethanolamine**; (GlcNAc)1 (Ino-P)1; (GlcN)1 (Ino(acyl)-P)1 (Man)2 (EtN)1 (P)1; (GlcN)1 (Ino(acyl)-P)1 (Man)1; (GlcN)1 (Ino-P)1; Palmityl-CoA; (GlcN)1 (Ino(acyl)-P)1 (Man)1 (EtN)1 (P)1; (GlcN)1 (Ino(acyl)-P)1 (Man)3 (EtN)2 (P)2; 6-(alpha-d-glucosaminyl)-1D-myo-inositol
	Sphingolipid Metabolism	Sphinganine; Ceramide 1-phosphate; **Sphingosine 1-phosphate**; Sphinganine 1-phosphate; Dihydroceramide; Phytoceramide; SM; Ceramide; Palmityl-CoA; l-Serine; 3-Dehydrosphinganine; Galabiosylceramide; Galactosylceramide; GM4; Lactosylceramide; Glucosylceramide; Sphingosine; 3-*O*-Sulfogalactosylceramide (d18:1/24:0); Phytosphingosine; Digalactosylceramidesulfate; *O*-Phosphoethanolamine
Day 17	**Linoleic Acid Metabolism**	**Linoleic acid**; **Phosphatidylcholine**; 9,10-Epoxyoctadecenoic acid; 12,13-EpOME; **13-l-Hydroperoxylinoleic acid**
	**Glycerophospholipid Metabolism**	**Phosphatidylethanolamine**; **Phosphatidylcholine**; Dihydroxyacetone phosphate; **LysoPC(18:1(9Z))**; 1,2-Diacyl-sn-glycerol; Citicoline; Phosphorylcholine; Choline; Acetylcholine ; *O*-Phosphoethanolamine; Ethanolamine; **PA(16:0/16:0)**; Acyl-CoA; 1-Acyl-sn-glycerol 3-phosphate; CDP-diacylglycerol; Glycerol 3-phosphate; 1-Acyl-sn-glycero-3-phosphoethanolamine; 2-Acyl-sn-glycero-3-phosphoethanolamine; 2-Acyl-sn-glycero-3-phosphocholine; CDP-glycerol; PS(16:0/16:0); Phosphatidylglycerol; Dihydroxyacetone Phosphate Acyl Ester; CDP-Ethanolamine; 1-Phosphatidyl-d-myo-inositol; Glycerylphosphorylethanolamine; Glycerophosphocholine; Phosphatidyl-*N*-methylethanolamine; Phosphatidylglycerophosphate; Cardiolipin
	**Sphingolipid Metabolism**	Sphinganine; Ceramide 1-phosphate; **Sphingosine 1-phosphate**; **Sphinganine 1-phosphate**; Dihydroceramide; Phytoceramide; **SM**; Ceramide; Palmityl-CoA; l-Serine; 3-Dehydrosphinganine; Galabiosylceramide; Galactosylceramide; GM4; Lactosylceramide; Glucosylceramide; Sphingosine; 3-*O*-Sulfogalactosylceramide (d18:1/24:0); Phytosphingosine; Digalactosylceramidesulfate; *O*-Phosphoethanolamine
	Biotin Metabolism	Biotin; Biotinyl-5′-AMP; Biocytin; Holo-[carboxylase]; **l-Lysine**
	Alpha-Linolenic Acid Metabolism	OPC4-CoA; OPC6-CoA; OPC8-CoA; **Phosphatidylcholine**; Alpha-Linolenic acid; *trans*-2-Enoyl-OPC4-CoA; t*rans*-2-Enoyl-OPC6-CoA; *trans*-2-Enoyl-OPC8-CoA; Stearidonic acid
Day 21	**Linoleic Acid Metabolism**	**Linoleic acid**; **Phosphatidylcholine**; 9,10-Epoxyoctadecenoic acid; 12,13-EpOME; 13-l-Hydroperoxylinoleic acid
	**Glycerophospholipid Metabolism**	**Phosphatidylethanolamine**; **Phosphatidylcholine**; Dihydroxyacetone phosphate; **LysoPC(18:1(9Z))**; 1,2-Diacyl-sn-glycerol; Citicoline; Phosphorylcholine; Choline; Acetylcholine ; *O*-Phosphoethanolamine; Ethanolamine; PA(16:0/16:0); Acyl-CoA; 1-Acyl-sn-glycerol 3-phosphate; CDP-diacylglycerol; Glycerol 3-phosphate; 1-Acyl-sn-glycero-3-phosphoethanolamine; 2-Acyl-sn-glycero-3-phosphoethanolamine; 2-Acyl-sn-glycero-3-phosphocholine; CDP-glycerol; PS(16:0/16:0); Phosphatidylglycerol; Dihydroxyacetone Phosphate Acyl Ester; CDP-Ethanolamine; 1-Phosphatidyl-d-myo-inositol; Glycerylphosphorylethanolamine; Glycerophosphocholine; Phosphatidyl-*N*-methylethanolamine; Phosphatidylglycerophosphate; Cardiolipin
	**Histidine Metabolism**	l-Glutamic acid; 4-Imidazolone-5-propionic acid; Urocanic acid; **l-Histidine**; Imidazole-4-acetaldehyde; 1-Methylhistamine; Methylimidazole acetaldehyde; **Histamine**; Carnosine; *N*-Formyl-l-aspartate; Formiminoglutamic acid; Imidazoleacetic acid; Methylimidazoleacetic acid; l-Aspartic acid; 1-Methylhistidine
	**Biosynthesis of Unsaturated Fatty Acids**	(13Z,16Z)-Docosadi-13,16-enoyl-CoA; Tetracosenoyl-CoA; Docosenoyl-CoA; Icosenoyl-CoA; Tetracosanoyl-CoA ; Docosanoyl-CoA; (7Z,10Z,13Z,16Z)-Docosatetraenoyl-CoA; (11Z,14Z)-Icosadienoyl-CoA; (7Z,10Z,13Z,16Z,19Z)-Docosapentaenoyl-CoA; (11Z,14Z,17Z)-Icosatrienoyl-CoA; Palmityl-CoA; Stearoyl-CoA; Eicosanoyl-CoA; Oleoyl-CoA; Linoleoyl-CoA; Arachidonyl-CoA; 8,11,14-Eicosatrienoyl-CoA; Gamma-linolenoyl-CoA; (4Z,7Z,10Z,13Z,16Z,19Z)-Docosahexaenoyl-CoA; (5Z,8Z,11Z,14Z,17Z)-Icosapentaenoyl-CoA; Alpha-Linolenoyl-CoA; 13,16-Docosadienoic acid; Nervonic acid; Erucic acid; Icosenoic acid; Tetracosanoic acid; Behenic acid; 7,10,13,16-Docosatetraenoic acid; Icosadienoic acid; Clupanodonic acid; Icosatrienoic acid; **Palmitic acid**; Stearic acid; Arachidic acid; Oleic acid; **Linoleic acid**; **Arachidonic acid**; 8,11,14-Eicosatrienoic acid; Gamma-Linolenic acid; (4Z,7Z,10Z,13Z,16Z,19Z)-Docosahexaenoic acid; Eicosapentaenoic acid; Alpha-Linolenic acid
	Sphingolipid Metabolism	Sphinganine; Ceramide 1-phosphate; **Sphingosine 1-phosphate**; Sphinganine 1-phosphate; Dihydroceramide; Phytoceramide; **SM**; Ceramide; Palmityl-CoA; L-Serine; 3-Dehydrosphinganine; Galabiosylceramide; Galactosylceramide; GM4; Lactosylceramide; Glucosylceramide; Sphingosine; 3-*O*-Sulfogalactosylceramide (d18:1/24:0); Phytosphingosine; Digalactosylceramidesulfate; *O*-Phosphoethanolamine

**Table 6 ijms-20-06348-t006:** Disease pathway analysis showing the top 5 disease associated pathways and their respective compounds involved across the exposure days. Disease pathways that are significantly different (*p* < 0.05) between PEPs-exposed and HEPA control rat serum are highlighted bold. Compounds that are significantly different (*p* < 0.05) in PEPs-exposed rat serum from HEPA control are in red text.

	Disease Pathway Analysis	Compound
Day 1	**Glycogenosis, Type VII. Tarui Disease**	**Bilirubin**; **Uric acid**
	Lesch-Nyhan Syndrome	Hypoxanthine; **Uric acid**; **Uridine**; Xanthine; Folic acid
	Liver Disease due to Cystic Fibrosis	Chenodeoxycholic acid; **Cholic acid**; **Deoxycholic acid**; Glycocholic acid; Lithocholic acid; Ursodeoxycholic acid
	2-Hydroxyglutaric Academia (l)	**l-Lysine**; l-2-Hydroxyglutaric acid
	Familial Lipoprotein Lipase Deficiency	Triglycerides; **Uric acid**
Day 5	**Hypertension**	Norepinephrine; Normetanephrine; Metanephrine; 20-Hydroxyeicosatetraenoic acid; Angiotensin II; Angiotensin III; **Alpha-Linolenic acid**; **Arachidonic acid**; **Docosahexaenoic acid**; Eicosapentaenoic acid; **Linoleic acid**; Dimethyl-l-arginine
	**Dengue Fever**	l-Alanine; **l-Histidine**; **l-Phenylalanine**
	**Gestational Diabetes Mellitus**	**Arachidonic acid**; **Docosahexaenoic acid**; *O*-Phosphoethanolamine
	Histidinemia	Histamine; **l-Histidine**
	Inflammatory Diseases	**l-Phenylalanine**; l-Tyrosine
Day 9	**Hartnup disease**	**l-Tryptophan**; **5-Hydroxyindoleacetic acid**
	Carnitine Palmitoyl Transferase Deficiency (II)	d-Glucose; **l-Palmitoylcarnitine**; Ammonia; **Oleoylcarnitine**; Myoglobin; Long chain acyl carnitines; **Oleoylcarnitine**; l-Acetylcarnitine
	Spastic Ataxia	**5-Hydroxyindoleacetic acid**; Homovanillic acid; **l-Tryptophan**; Thiamine; Vanylglycol
	Progressive Familial Intrahepatic Cholestasis	24,25-Dihydroxyvitamin D; 25-Hydroxyvitamin D2; **Vitamin A**; **Bilirubin**; Alpha-Tocopherol; Cholesterol
	Acute or Chronic Demyelinating polyneuropathies|Mild Cognitive Impairment|Neuroborreliosis	**24-Hydroxycholesterol**; 27-Hydroxycholesterol
Day 13	Cystic Fibrosis	**24,25-Dihydroxyvitamin D**; d-Glucose; Vitamin A; Alpha-Tocopherol
	Progressive Familial Intrahepatic Cholestasis	**24,25-Dihydroxyvitamin D**; 25-Hydroxyvitamin D2; Vitamin A; Bilirubin; Alpha-Tocopherol; Cholesterol
Day 17	2-Hydroxyglutaric Academia (l)	**l-Lysine**; l-2-Hydroxyglutaric acid
	Histidinemia	Histamine; **l-Histidine**
	Hyperlysinemia I, Familial Hyperpipecolatemia	Pipecolic acid; **l-Lysine**
	2,4-Dienoyl-CoA Reductase Deficiency	**l-Lysine**; Decadienoylcarnitine (C10:2); l-Carnitine
	Dengue Fever	l-Alanine; **l-Histidine**; l-Phenylalanine
Day 21	**Histidinemia**	**Histamine**; **l-Histidine**
	Alpha-1-Antitrypsin Deficiency (AATD)	alpha-1-Antitrypsin; **Glycocholic acid**
	Dengue Fever	l-Alanine; **l-Histidine**; l-Phenylalanine
	Gestational Diabetes Mellitus	**Arachidonic acid**; Docosahexaenoic acid; *O*-Phosphoethanolamine
	Nephrotic Syndrome	25-Hydroxyvitamin D2; Calcidiol; **Histamine**

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
