# Peer review of "Integrated Transcriptomics, Metabolomics, and Lipidomics Profiling in Rat Lung, Blood, and Serum for Assessment of Laser Printer-Emitted Nanoparticle Inhalation Exposure-Induced Disease Risks"

_ijms, 2019, doi:10.3390/ijms20246348_

Round 1

Reviewer 1 Report

Here the authors systematically studied integrating transcriptomics, metabolomics, and lipidomics in rat models, assessed PEPs inhalation exposure-induced disease risks. This study employed genome-wide mRNA and miRNA profiling integrated with metabolomics and lipidomics profiling in rat, and discovered important mRNAs, miRNAs, lipids and metabolites may serve as candidate biomarkers for future occupational and medical surveillance studies. This study is interesting and important, I have only one questions: Do these disorders in metabolomics and lipidomics profiling contribute to the clinical disorder? As many data shown on Day 21, such as in Fig3-4 were not changed.

Author Response

Reviewer 1:

Here the authors systematically studied integrating transcriptomics, metabolomics, and lipidomics in rat models, assessed PEPs inhalation exposure-induced disease risks. This study employed genome-wide mRNA and miRNA profiling integrated with metabolomics and lipidomics profiling in rat, and discovered important mRNAs, miRNAs, lipids and metabolites may serve as candidate biomarkers for future occupational and medical surveillance studies. This study is interesting and important, I have only one questions: Do these disorders in metabolomics and lipidomics profiling contribute to the clinical disorder? As many data shown on Day 21, such as in Fig3-4 were not changed.

Authors:

We thank the reviewer for the positive comments. The observed metabolites and lipids returned to baseline level at Day 21 in Fig 3-4. However, several metabolites and lipids varied significantly at multiple time points during the continuous PEPs inhalation exposure. For instance, histidine quantification is significantly lowered in day 5 (p=0.016) and day 17 (p=0.033) of continuous PEPs inhalation (Figure 3A), and although non-significant, a similar trend was observed with histamine (Figure 3B), a downstream product of histidine. High DHA levels associate with linoleic and arachidonic acid, suggesting an association with a resolving inflammatory response following PEPs inhalation. Inflammation peaked at Day 9 and was suppressed from Day 13 to Day 17 followed by high DHA expression which returned to baseline by Day 21 (Figure 2C). In clinics, histidine levels are low in patients with heart failure and are correlated with B-type natriuretic peptide, left ventricle ejection fraction, left ventricle end-diastolic volume index, inferior vena cava diameter and the ratio of early diastolic velocity of the mitral inflow to mitral annulus, respectively [1]. The variation of the studied metabolites and lipids is consistent with the observed adverse cardiovascular response in the same set of rats reported by our group [2]. A more detailed discussion on the clinical implications is now provided in lines 511-517.

References

Hakuno, D., et al., Plasma Amino Acid Profiling Identifies Specific Amino Acid Associations with Cardiovascular Function in Patients with Systolic Heart Failure. PLOS ONE, 2015. 10(2): p. e0117325. Alex P. Carll, R.S., Sandra V. Pirela, Yun Wang, Zhengzhi Xie, Pawel Lorkiewicz, Nazratan Naeem, Yong Qian, Vincent Castranova, John J. Godleski, Philip Demokritou, Inhalation of Printer-Emitted Particles Impairs Cardiac Conduction, Hemodynamics, and Autonomic Regulation and Induces Arrhythmia and Electrical Remodeling in Rats. Particle and Fibre Toxicology, 2019. in revision.

Reviewer 2 Report

This study assessed the potential risk of inhaling laser-printer emitted particles (PEPs) in causing pulmonary and cardiovascular diseases by genome-wide mRNA and miRNA profiling in rat model. It ties to our real life tightly because nano-particulates can be released in a lot of living scenarios. Numerous genes and compounds have been studied in different metabolic or disease pathways from Day 1 to Day 21.

It is not very clear about the distinction between pristine engineered nanoparticles (ENPs) and life cycle particulate matter (LCPM). Author can give some examples behind each term. Besides, it is suggested to merge the first and second paragraphs in the introduction. It seems few pulmonary/respiratory relevant pathways have been listed in all tables. Does it mean PEPs inhalation exposure barely cause any pulmonary malfunction? The entire study revealed PEPs induced gene and disease relevant compound changes over 21 days. It is curious if the author plans to harvest targeted organs/tissues to see if there is any histological damage taking place? Because several gene or compound variations may not cause substantial tissue/organ disease.

Author Response

We thank the reviewer for the constructive comments.

All pristine materials are ENPs. Life cycle specific particulate matter (LCPM) is the term we introduced to explain that pristine ENPs across their cycle and value chain can be released in the air but their properties differ due to their transformations. For instance, an ENP which will be used in a nano-enabled paint will be released during sanding and drilling but the released LCPM will have different physico-chemical properties compared to pristine ENP used in the paint.  The same applies in this case to the pristine metallic nanoparticles used to formulate the toner of laser printers. When those ENPs are released in the air they will have completely different physico-chemical properties and will include in their surface organic compounds that are not there for the pristine ENPs. These explanation is added in the introduction (lines 62-70), and the first two paragraphs are merged now.

Our group reported that PEPs exposure caused no damage in the histological and chemiluminescence analysis of lung and heart tissues in the in vivo animal studies [1]. However, PEPs exposure elevated sympathetic influence, impaired ventricular performance and repolarization, and caused hypertension and arrhythmia, suggesting increased cardiovascular risk [2].  The results of the global transcriptomic profiling in this study confirmed these pathological data. No significant perturbation in molecular pathways or processes linking to pulmonary diseases was identified in rat lung or blood following PEPs inhalation. In contrast, PEPs inhalation exposure dysregulated genes and pathways involved in cardiovascular malfunction at every observed time point in rat lung and several time points in rat blood. We did harvest tissues from brain, heart, liver, and kidney from the rats in the same experimental setting. Our plan is to validate the results from this study using qRT-PCR and protein assays in the rat heart tissues, and our group has an ongoing study to sequence blood samples collected from occupational workers in printing companies. This discussion is now provided in lines 333-341, and lines 667-669.

References

Pirela, S.V., et al., A 21-day sub-acute, whole-body inhalation exposure to printer-emitted engineered nanoparticles in rats: Exploring pulmonary and systemic effects. NanoImpact, 2019. 15: p. 100176. Alex P. Carll, R.S., Sandra V. Pirela, Yun Wang, Zhengzhi Xie, Pawel Lorkiewicz, Nazratan Naeem, Yong Qian, Vincent Castranova, John J. Godleski, Philip Demokritou, Inhalation of Printer-Emitted Particles Impairs Cardiac Conduction, Hemodynamics, and Autonomic Regulation and Induces Arrhythmia and Electrical Remodeling in Rats. Particle and Fibre Toxicology, 2019. in revision.